# Engineering of acyl ligase domain in non-ribosomal peptide synthetases to change fatty acid moieties of lipopeptides

Rina Aoki [1], Eri Kumagawa[2], Kazuaki Kamata[2], Hideo Ago [3], Naoki Sakai[3,6], Tomohisa Hasunuma [4,5], Naoaki Taoka[1], Yukari Ohta[2,7] & Shingo Kobayashi [1] ✉

Cyclic lipopeptides (CLPs) produced by the genus *Bacillus* are amphiphiles composed of hydrophilic amino acid and hydrophobic fatty acid moieties and are biosynthesised by non-ribosomal peptide synthetases (NRPSs). CLPs are produced as a mixture of homologues with different fatty acid moieties, whose length affects CLP activity. Iturin family lipopeptides are a family of CLPs comprising cyclic heptapeptides and β-amino fatty acids and have antimicrobial activity. There is little research on how the length of the fatty acid moiety of iturin family lipopeptides is determined. Here, we demonstrated that the acyl ligase (AL) domain determines the length of the fatty acid moiety in vivo. In addition, enzyme assays revealed how mutations in the substrate-binding pocket of the AL domain affected substrate specificity in vitro. Our findings have implications for the design of fatty acyl moieties for CLP synthesis using NRPS.

Members of the genus *Bacillus* produce various peptide antibiotics[1], some of which are cyclic lipopeptides (CLPs). CLPs are amphiphiles comprising hydrophilic amino acid and hydrophobic fatty acid moieties and are classified into three main families (surfactin, fengycin, and iturin) based on their structures[2]. Surfactin family lipopeptides are heptapeptides that combine with β-hydroxy fatty acids to form cyclic lactone ring structures (Supplementary Fig. 1a) and display haemolytic and antibacterial activities[3]. Fengycin family lipopeptides are lipodecapeptides with an internal lactone ring in the peptide moiety and a β-hydroxy fatty acid chain (Supplementary Fig. 1b) and display antimicrobial and low haemolytic activity[4]. Iturin family lipopeptides are heptapeptides that combine with β-amino fatty acids to form cyclic lactam ring structures (Fig. 1, Supplementary Fig. 1c) and display antimicrobial and haemolytic activities[5–7]. The fatty acid moieties vary in length and have branched structures (*n*-, *iso*-, and *anteiso*-); these homologues are usually produced as mixtures. The length of the fatty acid moiety affects CLP activity. For example, surfactin, which has longer fatty acid moiety, has higher haemolytic activity[3]. Iturin family lipopeptides (iturin and mycosubtilin), which have longer fatty acid moieties, have stronger antimicrobial activities[8–10].

CLPs, such as surfactin, fengycin, and iturin family lipopeptides, are biosynthesised by non-ribosomal peptide synthetases (NRPSs). NRPSs are

multimodular enzymes composed of various domains. The amino acid moieties of CLPs are biosynthesised by the condensation (C), adenylation (A), peptidyl carrier protein (PCP), epimerisation (E), and thioesterase (Te) domains[11]. The A domain adenylates specific amino acids based on substrate specificity. The adenylated amino acid forms a thioester with a 4-phosphopantetheine cofactor attached to the adjacent PCP domain. 4-phosphopantetheinyl transferase (PPTase) converts *apo*-PCP to *holo*-PCP by phosphopantetheinylating the PCP domain. C domains are divided into several functional subtypes: $^L C_L$, $^D C_L$, and starter C domain[12]. The $^L C_L$ and $^D C_L$ domains are responsible for the growth of the peptide chain. These domains catalyse peptide bond formation between the C-terminal amino acid of a growing peptide chain attached to the upstream PCP domain and an amino acid attached to the downstream PCP domain. The $^L C_L$ domain catalyses peptide bond formation between two L-amino acids. The $^D C_L$ domain, the C domain downstream of the E domain, catalyses peptide bond formation between a D-amino acid and an L-amino acid[13]. The E domain catalyses the epimerisation of an L-amino acid or the C-terminal L-amino acid of a growing peptide chain attached to the adjacent PCP domain into the D-configuration. Because the E domain provides a mixture of L- and D-amino acids, the $^D C_L$ domain is responsible for selecting the correct D-amino acid[11]. Peptide chain elongation is terminated by the Te domain, a domain

[1]Agri-Bio Research Center, Kaneka Corporation, Takasago, Hyogo, Japan. [2]Gunma University Center for Food Science and Wellness, Gunma University, Maebashi, Gunma, Japan. [3]RIKEN SPring-8 Center, Sayo-gun, Hyogo, Japan. [4]Graduate School of Science, Technology, and Innovation, Kobe University, Nada, Kobe, Japan. [5]Engineering Biology Research Center, Kobe University, Nada, Kobe, Japan. [6]Present address: Japan Synchrotron Radiation Research Institute, Sayo-gun, Hyogo, Japan. [7]Present address: Laboratory of Food Microbiology, Department of Life and Food Sciences, School of Life and Environmental Sciences, Azabu University, Sagamihara, Kanagawa, Japan. ✉e-mail: Shingo.Kobayashi@kaneka.co.jp

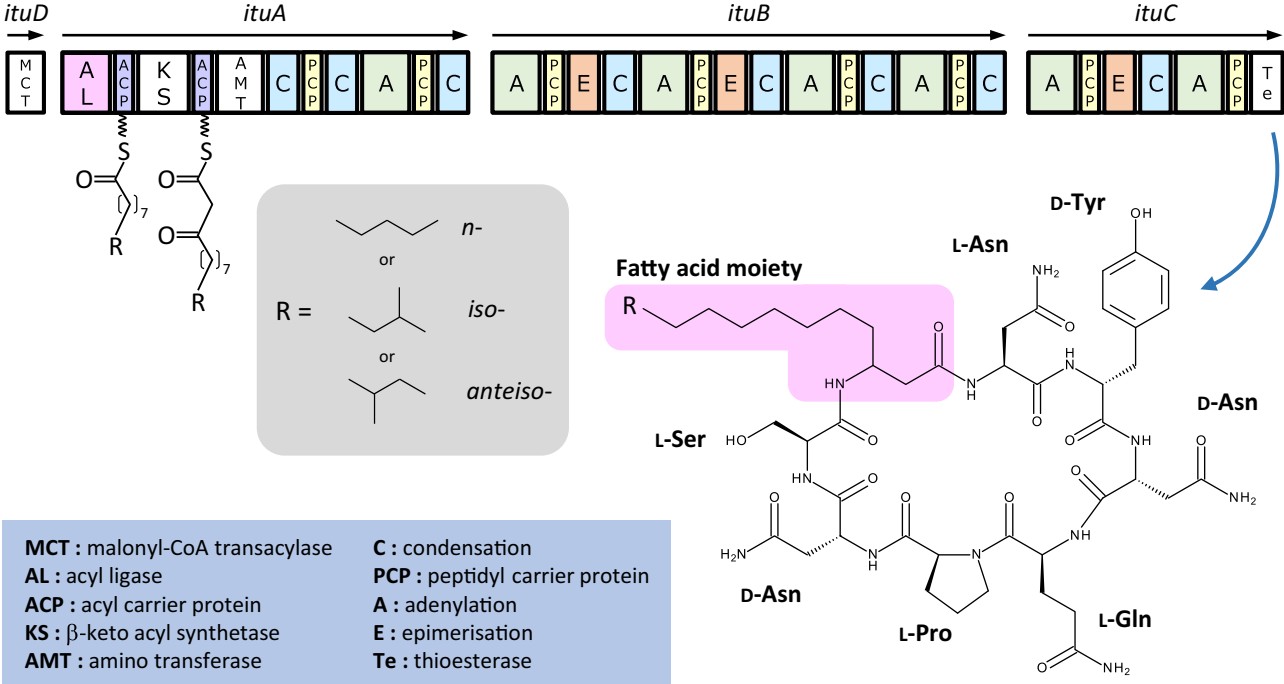

**Fig. 1 | Schematic representation of iturin synthetase and its product.** *ituD* encodes malonyl-CoA transacylase (MCT), and *ituA*, *ituB*, and *ituC* encode multiple modules. As an example of a product, the structure of an iturin homologue with a C15 fatty acid moiety is shown.

located at the end of NRPS, to release the product. The starter C domain, the C domain located at the beginning of NRPS, is responsible for transferring a fatty acid moiety to the first amino acid moiety. The fatty acid moieties of the surfactin and fengycin family lipopeptides (Supplementary Fig. 1a, b) are biosynthesised using β-hydroxy fatty acyl-CoA as substrates by the starter C domain[14,15]. Starter C domains have substrate specificity based on fatty acid length, and mutations to the starter C domain are known to alter its substrate specificity[16].

Iturin family lipopeptide synthetase utilises a fatty acid as its substrate to form a β-amino fatty acid moiety[17]. For example, for iturin synthetase (Fig. 1), the acyl ligase (AL) domain of ItuA converts fatty acids to fatty acyl-AMPs. Fatty acyl-AMP forms an acyl thioester with a 4-phosphopantetheine cofactor attached to the first acyl carrier protein (ACP) domain. PPTase converts *apo*-ACP to *holo*-ACP by phosphopantetheinylating the ACP domain. Malonyl-CoA transacylase (MCT), encoded by *ituD*, catalyses the transfer of a malonyl moiety from malonyl-CoA to a 4-phosphopantetheine cofactor attached to the second ACP domain of ItuA to form a malonyl thioester. The condensation of malonyl thioesters with acyl thioesters is then catalysed by the β-keto acyl synthetase (KS) domain to form β-keto acyl thioesters. In this step, the $C_n$ fatty acid, incorporated as a substrate, is elongated to form a $C_{n+2}$ β-keto acyl fatty acid. β-Keto acyl thioesters are converted to β-amino fatty acyl thioesters via transamination catalysed by the amino transferase (AMT) domain. Subsequently, the β-amino fatty acid is conjugated to the first PCP domain of ItuA by the activity of the first C domain of ItuA. An asparagine, which is activated by the A domain of ItuA, forms asparagine thioester with a 4-phosphopantetheine cofactor attached to the second PCP domain of ItuA. The β-amino fatty acid is then transferred to the asparagine thioester by the activity of the second C domain of ItuA. The β-amino fatty acyl asparagine is stepwise conjugated with six other amino acids through multiple domains [C, A, PCP, and E domains] of ItuB and ItuC, after which a linear peptide with a β-amino fatty acid is synthesised. Finally, the synthesised lipopeptide is cyclised by the Te domain to release iturin.

Iturin A and mycosubtilin are members of the iturin family lipopeptides that are biosynthesised as described above. Iturin A and mycosubtilin have amino acid moieties composed of L-Asn-D-Tyr-D-Asn-L-Gln-L-Pro-D-

Asn-L-Ser and L-Asn-D-Tyr-D-Asn-L-Gln-L-Pro-D-Ser-L-Asn, respectively (Fig. 1, Supplementary Fig. 1c). Iturin A produced by *Bacillus subtilis* RB14 (hereafter referred to as RB14) is mainly composed of C14, C15, and C16 homologues[18], whereas mycosubtilin produced by *B. subtilis* ATCC6633 (hereafter referred to as ATCC6633) is mainly composed of C16 and C17 homologues[19]. In contrast, the constituent fatty acids of *Bacillus* species are mainly composed of C15[20,21]. In iturin family lipopeptide synthetases, fatty acids are taken up by the AL domain and then elongated by two carbons by adjacent KS domains (Fig. 1). If iturin family lipopeptide synthetases have broad specificity based on fatty acid length, iturin family lipopeptides should be mainly composed of the C17 homologue derived from C15 fatty acid. However, the relative abundance of iturin family lipopeptide homologues produced by each *Bacillus* species differs. Therefore, the iturin family lipopeptide synthetases are likely to have selectivity for fatty acid substrates. A previous report suggested that AL and/or adjacent ACP domains are involved in substrate specificity[22]. However, whether these specificities directly determine the length of the fatty acid moiety in the CLP has not been demonstrated.

In this study, we aimed to identify the domain that determines the length of the fatty acid moiety of iturin family lipopeptides. We generated strains harbouring mutated AL and/or ACP domains and confirmed their effects in vivo. In addition, we investigated the effects of an amino acid substitution in the AL domain, which were found to significantly affect the fatty acid chain composition of the iturin family lipopeptides, on substrate specificity in vitro and on 3D structure in silico. Our findings provide useful information for designing the fatty acid moieties of lipopeptides produced by NRPS and greatly advance the technology for customising the activity of lipopeptides.

## Results
### Domains determining the length of the fatty acid moiety
In iturin family lipopeptide synthetases, fatty acids are adenylated by the AL domain, and fatty acyl-AMPs are transferred to the adjacent phosphopantetheinylated ACP domain to produce acyl thioesters[22] (Fig. 1). However, little information is available on how the length of the fatty acid moiety of iturin family lipopeptides is determined. Accordingly, AL and/or ACP

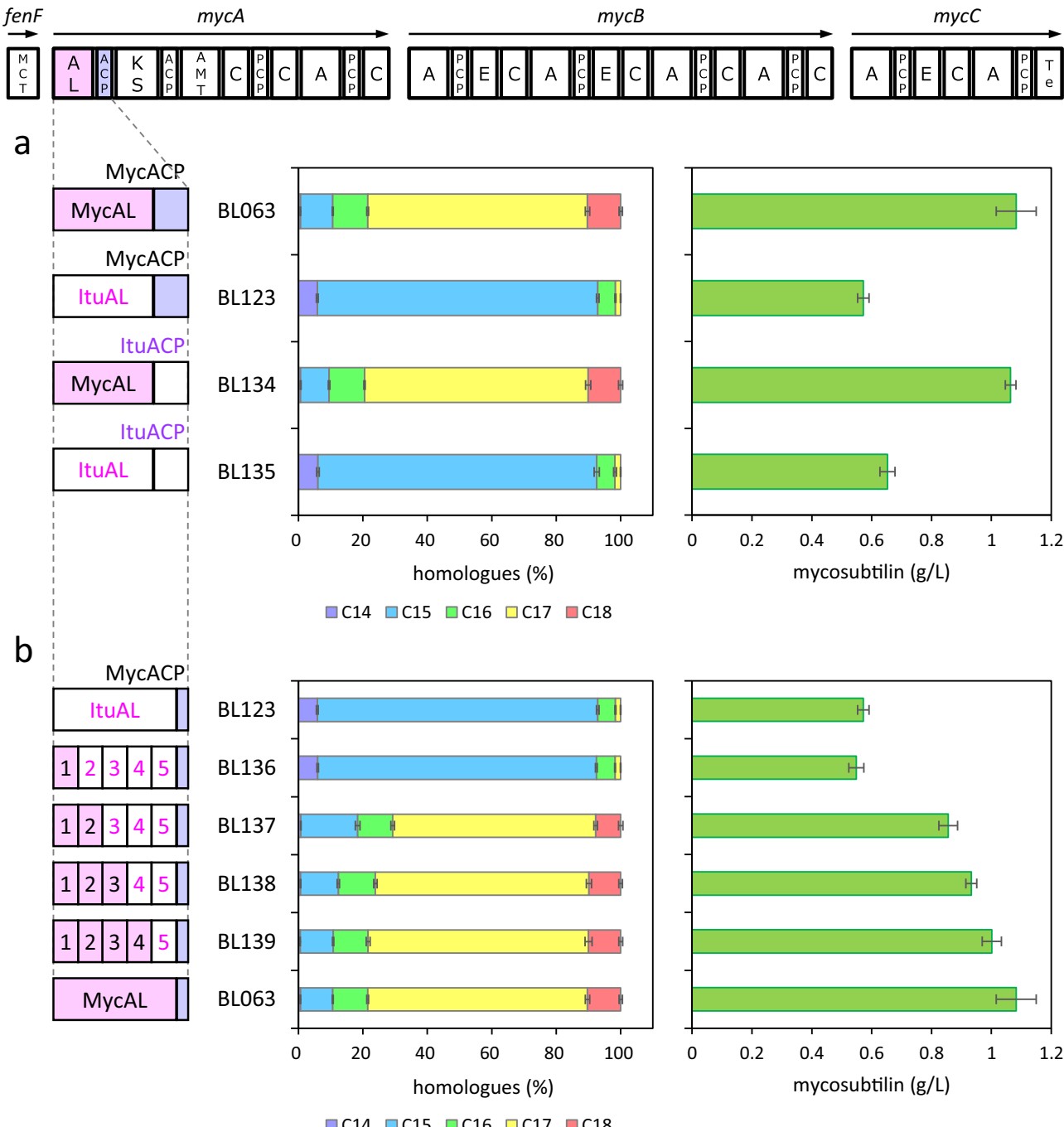

**Fig. 2 | Effect of acyl ligase (AL) and acyl carrier protein (ACP) domains on mycosubtilin production.** The relative abundance and productivity of mycosubtilin. Values and error bars represent the mean and the standard deviation, respectively, of four independent experiments. **a** AL and/or ACP substitution mutant strains. **b** Mutant strains with chimaeric AL domains.

domain substitution experiments were performed to identify the domains involved in determining the length of the fatty acid moiety in vivo.

First, we constructed BL063, which harbours the mycosubtilin synthetase operon derived from ATCC6633 under the *rplGB* promoter. Next, we constructed BL063-based mutant strains wherein the AL (MycAL) and/ or ACP (MycACP) domains of mycosubtilin synthetase were replaced by the AL (ItuAL) and/or ACP (ItuACP) domains of iturin synthetase from RB14. The amino acid sequences of ItuAL and MycAL were 84% identical, and those of ItuACP and MycACP were 74% identical (Supplementary Figs. 2, 3). The mutant strains were cultured to determine the relative abundance of mycosubtilin homologues. The mutant strains harbouring

MycAL and ItuAL produced C17 and C15 homologues as their major products, respectively (Fig. 2a). We found that $68 \pm 1\%$ of the mycosubtilins produced by BL063 were C17 homologues. The relative abundance of the mycosubtilin homologues produced by BL134 harbouring MycAL-ItuACP was similar to that of BL063. In contrast, $87 \pm 0.4$ and $87 \pm 1\%$ of the mycosubtilins produced by BL123 harbouring ItuAL-MycACP and BL135 harbouring ItuAL-ItuACP, respectively, were C15 homologues. These results clearly demonstrate that the AL domain, but not the ACP domain, is responsible for the relative abundance of mycosubtilin homologues.

The mutant strains harbouring MycAL showed higher mycosubtilin productivity than the strains harbouring ItuAL. The mycosubtilin

productivity of BL134 (1.1 ± 0.02 g/L) was comparable to that of BL063 (1.1 ± 0.1 g/L) (Fig. 2a). However, the mycosubtilin productivities of BL123 (0.6 ± 0.02 g/L) and BL135 (0.7 ± 0.03 g/L) were lower than that of BL063.

## Amino acid residues involved in determining the length of the fatty acid moiety

A comparison of the amino acid sequences of MycAL from ATCC6633 (585 amino acids) and ItuAL from RB14 (584 amino acids) revealed their considerable similarity (Supplementary Fig. 2). The specific amino acid residues that differ between MycAL and ItuAL were expected to determine the relative abundance of mycosubtilin homologues. The AL domain of iturin family lipopeptide synthetases is classified as fatty acyl-AMP ligase (FAAL)[22]. To determine the amino acid residues, structural modelling is usually employed. However, this strategy is not favourable because amino acid sequences of MycAL and ItuAL are less than 30% identical to those of FAALs[23–25] whose crystal structures have already been solved. Therefore, we experimentally verified all amino acid residues that are different between MycAL and ItuAL.

To narrow down the regions affecting the relative abundance of mycosubtilin homologues, we constructed mutant strains harbouring MycAL and ItuAL chimaeras. The AL domain was divided into five regions, each containing approximately 100 amino acids (Supplementary Fig. 2), and four types of chimaeras were constructed (Fig. 2b, Table 1). The boundary of the region was set at the part where the amino acid sequence was identical in the two sequences to prevent the decrease in activity of the chimaeric AL domains. Because multiple amino acids may be involved in chain length determination, we successively replaced the ituAL sequence with the MycAL sequence. The relative abundances of the mycosubtilin homologues produced by these mutant strains are shown in Fig. 2b.

Upon the replacement of the region 2 of ItuAL by that of MycAL, C17 homologue became the major product instead of C15 homologue. BL136, in which region 1 of ItuAL was replaced with that of MycAL, mainly produced C15 homologues that were almost similar to those produced by BL123, wherein the full-length AL domain was ItuAL. In contrast, the relative abundances of the C15 homologues produced by BL137 (a mutant strain in which regions 1 and 2 of ItuAL were replaced with those of MycAL), BL138 (a mutant strain in which regions 1, 2, and 3 of ItuAL were replaced with those of MycAL), and BL139 (a mutant strain in which regions 1, 2, 3, and 4 of ItuAL were replaced with those of MycAL) were 18 ± 1, 12 ± 0.4, and 10 ± 0.2%, respectively. The relative abundances of the C17 homologues produced by BL137, BL138, and BL139 were 63 ± 1, 66 ± 1, and 68 ± 1%, respectively. The relative abundance of mycosubtilin homologues produced by BL139 was comparable to that of BL063, in which the full-length AL domain was MycAL. These results suggest that the amino acid residues in regions 2, 3, and 4 determined fatty acid moiety length. Mycosubtilin productivity gradually improved in BL137, BL138, and BL139, and the productivities of BL139 (1.0 ± 0.03 g/L) and BL063 (1.1 ± 0.1 g/L) were comparable (Fig. 2b).

To identify the amino acid residues in regions 2, 3, and 4 that determine the length of the fatty acid moiety, we generated mutant strains containing ItuAL with an amino acid substitution. For example, the 111th amino acid of ItuAL is His, and the corresponding amino acid residue of MycAL is Asn (Supplementary Fig. 2). Therefore, we generated a strain containing a mutant ItuAL in which the 111th His was substituted by Asn and named it H111N. According to this rule, we generated 48 mutant strains containing ItuAL with an amino acid substitution mutation and confirmed the relative abundance of mycosubtilin homologues produced by these mutant strains (Supplementary Figs. 4, 5).

Among all mutant strains, the F208L mutant strain showed a clear increase in the production of C17 and C18 homologues. In region 2, 26% and 83–87% of the mycosubtilins produced by the F208L and other mutant strains, respectively, were C15 homologues. In region 3, 79% and 82–86% of the mycosubtilins produced by the K287R and other mutant strains, respectively, were C15 homologues. In region 4, 83% and 87–89% of the mycosubtilins produced by the A332T and other mutant strains,

respectively, were C15 homologues. These results suggest that the F208L mutation in region 2, K287R mutation in region 3, and A332T mutation in region 4 caused changes in the relative abundance of mycosubtilin homologues. Among these, F208 had a particularly obvious effect on abundance. The F208 residue is highly conserved in the AL domain of iturin family lipopeptide synthetases, except for the mycosubtilin synthetase derived from ATCC6633 in which the corresponding amino acid residue is Leu (Supplementary Fig. 6).

Subsequently, we investigated whether the relative abundance of iturin homologues was altered when these substitutions were introduced in iturin synthetase. Firstly, we generated the iturin-producing mutant strain BL127, which harbours the iturin synthetase operon derived from RB14 under the *rplGB* promoter. Iturin production by BL127 amounted to 0.9 ± 0.1 g/L (Fig. 3a). Of the iturins produced by BL127, 82 ± 2% were C15 homologues (Fig. 3a).

Secondly, we generated mutant strains with amino acid substitution(s) in ItuAL. The mutant strains with F208L mutation showed a higher production ratio of C17 homologue, and the C17 production ratio was increased further by adding K287R and A332T mutations. As shown in Fig. 3a, the relative abundances of C15 homologues produced by mutant strains with single mutations in F208L, K287R, and A332T were 29 ± 0.2, 72 ± 0.4, and 75 ± 2%, respectively. In contrast, the relative abundances of the C17 homologues produced by BL127, F208L, K287R, and A332T mutant strains were 3 ± 0.2, 53 ± 0.3, 11 ± 0.2, and 6 ± 0.3%, respectively. The relative abundances of C17 homologues produced by the F208L and K287R double mutant strain and the F208L, K287R, and A332T triple mutant strain were 66 ± 1 and 69 ± 1%, respectively. The relative abundance of the C17 homologues produced by the triple mutant strain was similar to that produced by BL128 (75 ± 1%), in which MycAL replaced the AL domain. The synergistic effects of mutational duplication were observed for the relative abundance of iturin homologues. The iturin productivity of the F208L, K287R, and A332T triple mutant strain (1.8 ± 0.1 g/L) was higher than that of BL127 and comparable to that of BL128 (1.9 ± 0.1 g/L) (Fig. 3a).

## Saturation mutagenesis of F208

We generated ItuAL mutant strains in which the Phe residue at position 208, which had the greatest influence on the relative abundance of iturin homologues, was replaced with a proteinogenic amino acid other than Phe and confirmed the relative abundance of iturin homologues produced by these mutant strains (Fig. 3b). The number of atoms in the side chain that are replaced and the relative abundance of iturin homologues produced were likely correlated. Upon replacement with amino acid residues whose number of side chain atoms was less than or equal to three, homologues with C17 or larger fatty acids occupied a major fraction of the product. The maximum fraction of homologues with C17 or larger fatty acids observed in F208S reached 91% of the total product. The replacement of amino acid residues whose number of side chain atoms was larger than three affected the fraction of homologues with C17 or larger fatty acids. The mutations of F208D, F208N, F208I, F208E, and F208H maintained a higher fraction of homologues with C17 or larger fatty acids, whereas F208M, F208L, F208Q, and F208K showed a decrease in the fraction of homologues with C17 or larger fatty acids. In cases of amino acid residues with four side chain atoms, the bulky portion existing at the distal end of its side chain from its main chain, i.e., the sulphur atom of Met or the isobutane group of Leu, seemed to inhibit the maintenance of homologues with C17 or larger fatty acids. Replacement by an amino acid residue with seven or more side chain atoms reduced critically total amount of iturin homologues, although there was a possibility that the amino acid residue with a large side chain decreased the structural integrity of mutant enzymes. These results strongly suggest that the size of the amino acid at position 208 plays a crucial role in the substrate specificity of the AL domain.

The mutant strains F208G, F208A, F208C, F208S, F208P, F208T, F208D, F208N, F208E, and F208H produced C19 homologues (Fig. 3b and Supplementary Tables 1, 2), which were not produced in the F208L mutant strain. In particular, the relative abundance of C19 homologues in the

## Table 1 | Strains and plasmids used in this study

| Strain or plasmid | Genotype or description | Reference or source |
|---|---|---|
| **Escherichia coli** | | |
| E. coli JM109 | recA1, endA1, gyrA96, thi-1, hsdR17($r_K^-$ $m_K^+$), e14$^-$ (mcrA$^-$), supE44, relA1, Δ(lac-proAB)/ F' [traD36, proAB$^+$, lac I$^q$, lacZΔM15] | Takara Bio Inc. |
| E. coli BL21-AI | F$^-$, ompT, hsdS$_B$ ($r_B^-$ $m_B^-$), gal, dcm, araB::T7RNAP-tetA | Thermo Fisher Scientific |
| **Bacillus subtilis** | | |
| B. subtilis 168 | trpC2 | Lab stock |
| B. subtilis ATCC6633 | Wild-type (NBRC 3134) | NITE Biological Resource Center |
| B. subtilis RB14 | Wild-type | 45, 50 |
| KB04 | ΔppsABCDE::fenF-mycA-mycB-mycC-kan, sfp::lpa-14-cat | 40 |
| BL043 | ΔppsABCDE::erm-PrplGB-fenF-mycA-mycB-mycC-kan, sfp::lpa-14-cat | This work |
| BL063 | BL043 ΔsigF | This work |
| BL082 | BL063 ΔmycAL-mycACP::pheS*-spec | This work |
| BL123 | BL063 ΔmycAL::ituAL | This work |
| BL134 | BL063 ΔmycACP::ituACP | This work |
| BL135 | BL063 ΔmycAL-mycACP::ituAL-ituACP | This work |
| BL136 | BL063 ΔmycAL::chimaeric AL domain 1 (region 1 derived from mycAL and regions 2, 3, 4, and 5 derived from ituAL) | This work |
| BL137 | BL063 ΔmycAL::chimaeric AL domain 2 (regions 1 and 2 derived from mycAL and regions 3, 4, and 5 derived from ituAL) | This work |
| BL138 | BL063 ΔmycAL::chimaeric AL domain 3 (regions 1, 2, and 3 derived from mycAL and regions 4 and 5 derived from ituAL) | This work |
| BL139 | BL063 ΔmycAL::chimaeric AL domain 4 (regions 1, 2, 3, and 4 derived from mycAL and region 5 derived from ituAL) | This work |
| BL083 | BL063 Δerm-PrplGB-fenF-mycAL-mycACP::spec | This work |
| XyyyZ | BL063 ΔmycAL::ituAL_XyyyZ | This work |
| BL113 | ΔppsABCDE::spec, sfp::lpa-14-cat, ΔsigF | This work |
| BL144 | ΔppsABCDE::spec-ituA'-ituB-ituC, sfp::lpa-14-cat, ΔsigF, srfAB::kan | This work |
| BL127 | ΔppsABCDE::erm-PrplGB-ituD-ituA-ituB-ituC, sfp::lpa-14-cat, ΔsigF, srfAB::kan | This work |
| BL128 | BL127 ΔituAL::mycAL | This work |
| F208L | BL127 ΔituAL::ituAL_F208L | This work |
| K287R | BL127 ΔituAL::ituAL_K287R | This work |
| A332T | BL127 ΔituAL::ituAL_A332T | This work |
| F208L, K287R | BL127 ΔituAL::ituAL_F208L, K287R | This work |
| F208L, K287R, A332T | BL127 ΔituAL::ituAL_F208L, K287R, A332T | This work |
| F208X | BL127 ΔituAL::ituAL_F208X | This work |
| MycAL_L208X | BL127 ΔituAL::mycAL_L208X | This work |
| **Fungi** | | |
| Rhizoctonia solani | Wild-type (MAFF 242303) | Genebank Project, NARO |
| **Yeast** | | |
| Saccharomyces cerevisiae BY4742 | MATalpha his3Δ1 leu2Δ0 lys2Δ0 ura3Δ0 | Horizon Discovery, Ltd. |
| **Plasmids** | | |
| pUC19 | Cloning vector | New England Biolabs |
| pUC19-mycU-erm-PrplGB-fenF-ituAL-mycACP-mycAD | QuikChange template plasmid for generating XyyyZ strains | This work |
| pUC19-mycU-erm-PrplGB-ituD-ituAL-ituACP-ituAD | Construction of BL127 QuikChange template plasmid for generating F208X strains | This work |
| pUC19-mycU-erm-PrplGB-ituD-mycAL-ituACP-ituAD | Construction of BL128 QuikChange template plasmid for generating MycAL_L208X strains | This work |
| pET28a | Cloning vector for protein expression | Merck |
| pET28a-ituAL-ACP1 | Expression vector for His-tagged ItuAL_WT-MycACP | This work |
| pET28a-ituAL_F208G-ACP1 | Expression vector for His-tagged ItuAL_F208G-MycACP | This work |

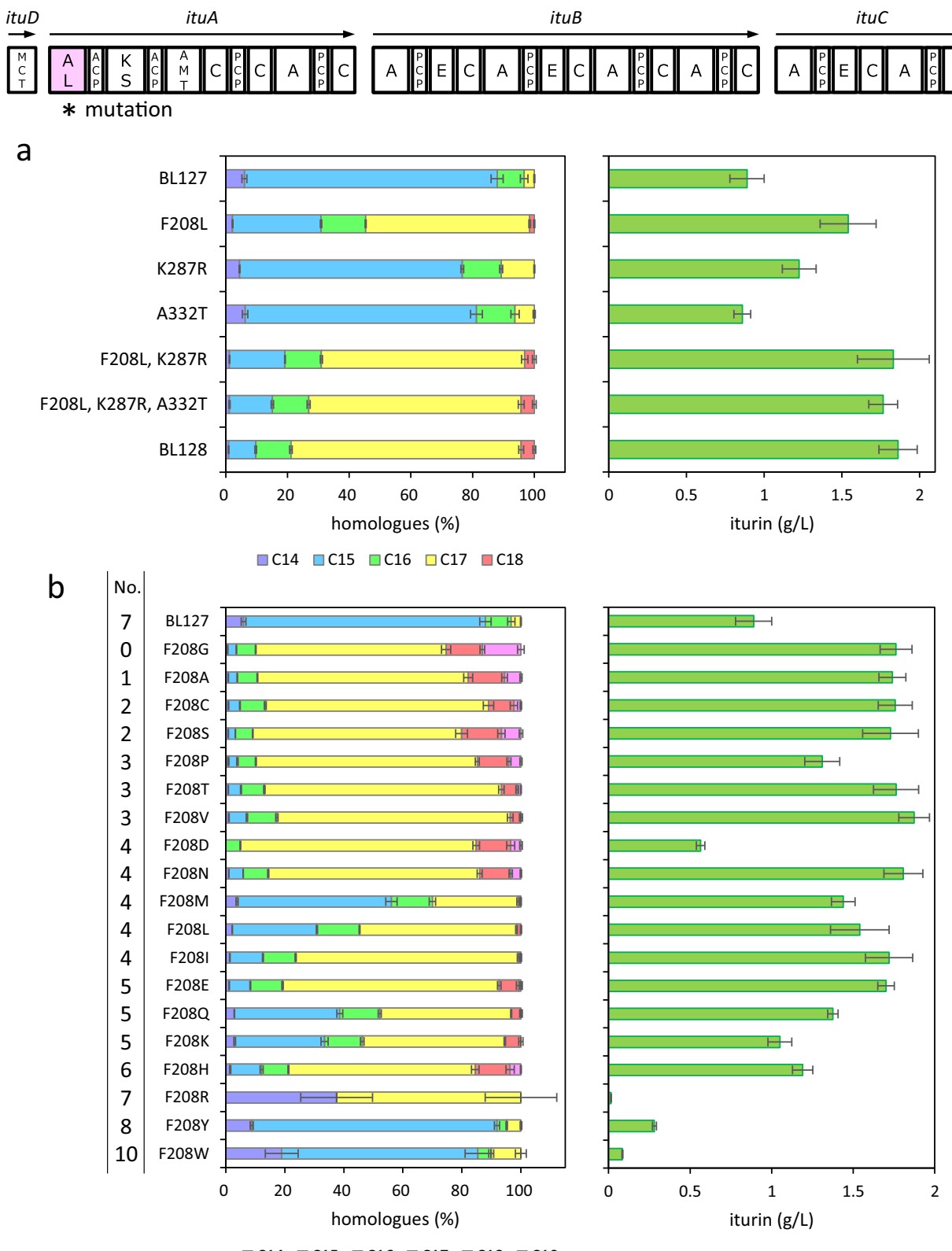

**Fig. 3 | Effect of amino acid substitutions in the acyl ligase (AL) domain on iturin production.** The relative abundance and productivity of iturin. Values and error bars represent the mean and the standard deviation, respectively, of four independent experiments. **a** Mutant strains containing F208L, K287R, and/or A332T mutation(s). **b** Mutant strains with mutations at F208. The number of atoms in the amino acid side chain is listed to the left of the strain name.

F208G mutant strain was the highest among the strains. In addition, we generated MycAL mutant strains in which the Leu residue at position 208 was replaced with Gly, Ala, Cys, Ser, Pro, Thr, Asp, Asn, Glu, or His and confirmed the relative abundance of iturin homologues produced by these mutant strains (Supplementary Fig. 7). Similar to ItuAL mutant strains, MycAL mutant strains produced C19 homologues.

In LC-TOF-MS analysis of iturin produced by the F208G mutant strain (Supplementary Table 1), a peak corresponding to the C14 homologue and

two peaks each corresponding to C15, C16, C17, C18, and C19 homologues were detected. These two peaks were assumed to correspond to homologues with *n*- and *iso*-fatty acid moieties for homologues with even-numbered fatty acid moieties, and to homologues with *iso*- and *anteiso*-fatty acids moieties for homologues with odd-numbered fatty acid moieties, respectively[2,8,10,18].

We performed lipid analysis of the F208G mutant strain cultured in a production medium (Supplementary Table 3). Since the production medium contains soybean flour, precipitates of the culture broth contain cells and soybean flour. Therefore, a production medium without inoculation of the strain was analysed in the same way. C15, C16, C17, and C18 fatty acids were detected in precipitates of the culture broth of the F208G mutant strain. However, only a small amount of C14 fatty acid was detected (0.2 ± 0.02%), and fatty acids with chain length less than C13 were not detected. *n*-C16 and *n*-C18 fatty acids were detected in precipitates of a production medium without inoculation of the strain. Since *n*-C16 and *n*-C18 fatty acids may be derived from soybean flour, it was suggested that at least *iso*-C14, *iso*-C15, *anteiso*-C15, *iso*-C16, *iso*-C17, and *anteiso*-C17 fatty acids were derived from the F208G mutant strain. Among these fatty acids, *anteiso*-C15 was the most abundant (11.6 ± 0.6%).

## Measurement of acyl ligase activity using the AL domain

The AL activity of iturin family lipopeptide synthetases was determined by Hansen et al. using a recombinant protein with mycosubtilin synthetase AL and adjacent ACP domains in vitro[22]. Therefore, fusion proteins of the wild-type iturin synthetase AL domain (ItuAL_WT) or the F208G mutant (ItuAL_F208G) with the mycosubtilin synthetase ACP domain were obtained as N-terminally His-tagged proteins, and relative acyl ligase activity was measured using *n*-fatty acids of various chain lengths as substrates. In our assay, fatty acyl-AMPs were detected as products because the ACP domain was in the *apo*-form. As shown in Fig. 4a, ItuAL_WT exhibited the highest activity against lauric acid (C12), whereas ItuAL_F208G exhibited the highest activity against pentadecanoic acid (C15). In iturin family lipopeptide synthetases, fatty acids are taken up by the AL domain and then elongated by two carbons by adjacent KS domains (Fig. 1). This means that the C12 and C15 fatty acids incorporated as substrates correspond to the products with the C14 and C17 fatty acid moieties, respectively.

Subsequently, their activities against branched-chain fatty acids were evaluated. As previously reported[8,18], iturin contains *iso*- or *anteiso*-fatty acids as side chains, likely because of the composition of the constituent fatty acids of *Bacillus* species. When *n*-C15, *iso*-C15, and *anteiso*-C15 fatty acids were used as substrates, both ItuAL_WT and ItuAL_F208G utilised them. ItuAL_F208G was more active than ItuAL_WT for all C15 isomers (Fig. 4b).

## Predicted structure of the AL domain

To determine the relationship between the amino acid substitutions and changes in substrate specificity, the 3D structure of ItuAL was predicted from its amino acid sequence using ColabFold (Supplementary Fig. 8). Among the three residues identified as mediators of substrate specificity, F208 was predicted to constitute the substrate-binding pocket in the predicted structure of ItuAL_WT (Fig. 4c). In addition, F208 was located at the bottom of the pocket, suggesting that it may have determined substrate-binding pocket depth. In contrast, K287 and A332 were not involved in the formation of the substrate-binding pocket but were presumed to be located near the amino acid residues that form the pocket (Supplementary Fig. 9). Therefore, these residues may indirectly affect pocket shape. In the putative structure of ItuAL_F208G, substrate-binding pocket depth was increased (Fig. 4d), suggesting that long-chain fatty acids could be incorporated into the pocket. Similar changes in the depth of the substrate-binding pocket were observed in a comparison of the predicted structures of MycAL_WT and MycAL_L208G (Supplementary Fig. 10).

## Activities of iturins with different fatty acid chain compositions

Iturin and mycosubtilin, with longer side chains, are reported to have higher antimicrobial activity[8–10]. Therefore, we examined the activity of iturin mixtures. We extracted iturin from the culture broths of BL127, BL128, and F208G mutant strains and designated them C15-ITU, C17-ITU, and C19-ITU, respectively (Fig. 5a). The iturin extracts were then investigated for their antifungal activities against *Saccharomyces cerevisiae* (Fig. 5b). The results showed that C15-ITU had a minimal inhibitory concentration against *S. cerevisiae* of >40 mg/L, whereas those of C17-ITU and C19-ITU were 5 and 2.5 mg/L, respectively, indicating that iturin with a longer fatty acid moiety had higher antifungal activity, consistent with previous findings. In addition, the antifungal activities of C15-ITU and C19-ITU against the pathogenic plant fungi *Rhizoctonia solani* were compared. As shown in Fig. 5c, C19-ITU had a larger zone of inhibition against *R. solani* than that of C15-ITU, indicating higher antifungal activity.

Lipopeptides such as iturin are known to exhibit haemolytic activity. To ascertain whether haemolytic activity varied with fatty acid moiety length, C15-ITU and C19-ITU were used to determine haemolytic activity in Columbia sheep blood medium (Fig. 5d). The results showed that a clear haemolytic zone was formed with 2000 mg/L C19-ITU, but only slightly with C15-ITU, indicating that C19-ITU has higher haemolytic activity than that of C15-ITU.

## Discussion

In this study, we demonstrated that the AL domain, not the ACP domain, determines the relative abundance of iturin family lipopeptide homologues in vivo. In addition, we showed that an amino acid substitution in the AL domain alters substrate specificity in vitro. A previous report suggested that AL and/or adjacent ACP domains mediate substrate specificity[22]. Hansen et al. conducted a chase experiment using the *holo*-AL-ACP recombinant protein of mycosubtilin synthetase and showed that trans-2-decenoic (C10), decanoic (C10), myristic (C14), and palmitic acids (C16) were adenylated by the AL domain and then bound to the ACP domain, whereas acetic (C2), propionic (C3), butyric (C4), and hexanoic acids (C6) were not[22]. The chain length of the fatty acid loaded into the AL domain as a substrate approximately corresponded to the length of the mycosubtilin fatty acid moiety[10]. Here, we showed that *apo*-AL-ACP recombinant protein utilises odd-numbered (C11, C13, and C15) and branched-chain fatty acids (*iso*- and *anteiso*-C15) as substrates in addition to even-numbered (C12, C14, and C16) and straight-chain fatty acids which were reported by Hansen et al.[22]. Our results provide a more detailed characterisation of the AL domains that mediate specificity. Furthermore, we demonstrated that the substrate specificity of the AL domain directly determined the relative abundance of lipopeptide homologues. Previous studies have shown that growth temperature and medium composition affect the relative abundance of iturin family lipopeptide homologues. Fickers et al. revealed that the relative abundance of C15 and C17 mycosubtilin homologues produced by ATCC6633 increased at low growth temperatures, whereas that of C16 homologues decreased[19]. The addition of isoleucine to the medium stimulates the production of anteiso-C17 mycosubtilin homologues[26]. We have shown that modifying the AL domain can be a technique for providing lipopeptide compositions with different abundances of fatty acid moieties. The combination of AL domain modification and culture conditions would allow the production of more diverse lipopeptide compositions.

The mechanism of substrate recognition by the AL domain of iturin synthetase is similar to that of FAALs. The AL domain of iturin family lipopeptide synthetases is classified as FAAL[22]. In a previous study, amino acid residues in the substrate-binding pocket were found to play important roles in FAAL substrate specificity[27]. A bulky residue lining the binding pocket is reported to define the chain length specificity of the adenylate-forming enzyme superfamily (ANL superfamily) of enzymes, including FAALs[28]. The deeper the substrate-binding pocket, the longer the fatty acids it can accommodate. We conducted saturated mutation experiments at the

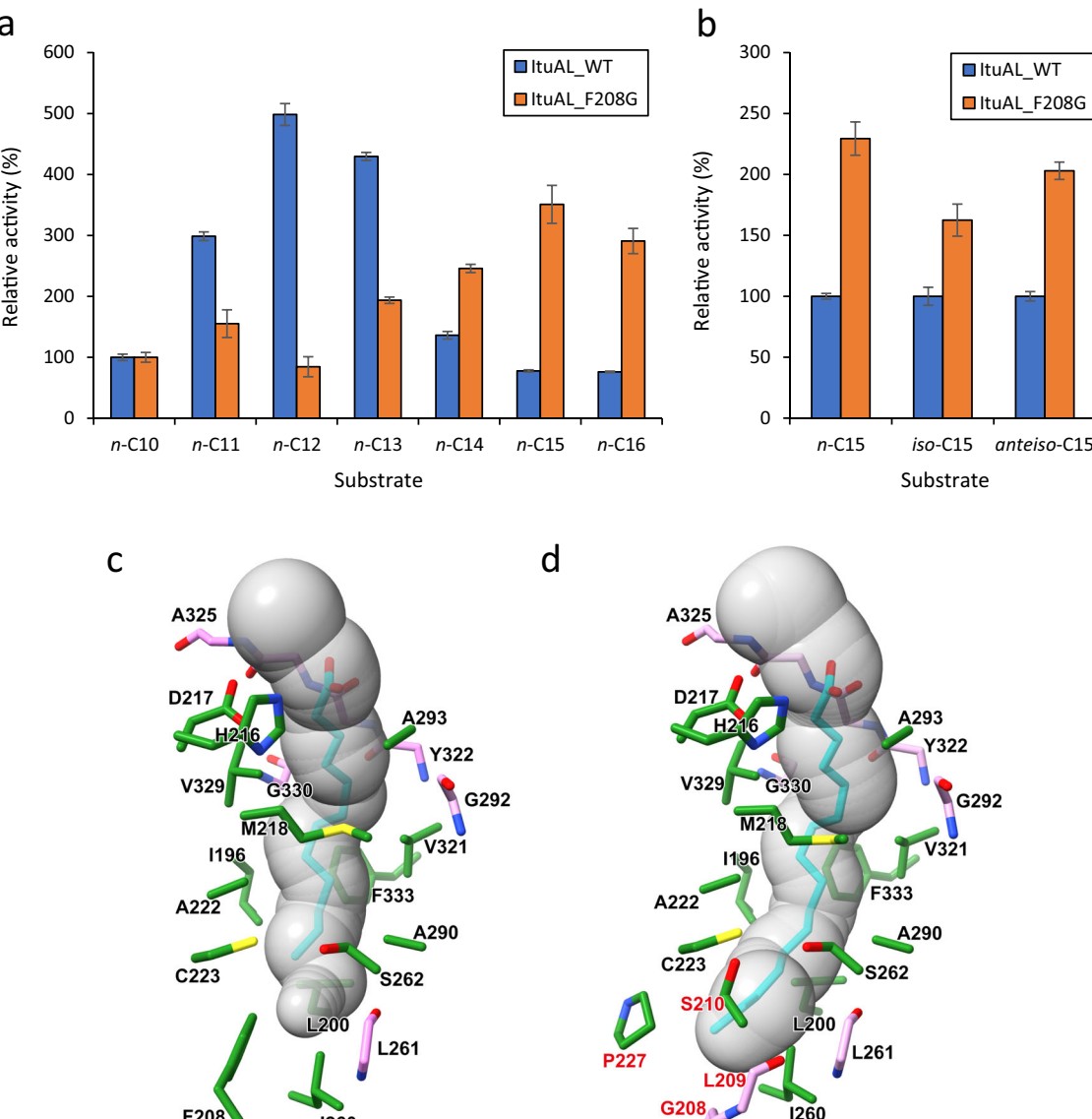

**Fig. 4 | Changes caused by F208G mutation. a** Relative activity of ItuAL_WT or ItuAL_F208G using each substrate. The activity of each enzyme using C10 as a substrate was defined as 100%. Values and error bars represent the mean and standard deviation, respectively, of three independent experiments. **b** Relative activity of ItuAL_F208G using each substrate. The activity of ItuAL_WT using the same substrate as that for ItuAL_F208G was defined as 100%. Values and error bars represent the mean and standard deviation, respectively, of three independent experiments. **c, d** Amino acid residues around the substrate-binding pocket in the predicted structures of ItuAL_WT (**c**) and ItuAL_F208G (**d**). The substrate-binding pocket detected using MOLEonline[51] is shown by connected grey-coloured spheres. The modelled fatty acid in the substrate-binding pocket is cyan-coloured. Amino acid residues are coloured as follows: amino acid residues indicated by green carbons are those whose side chains are involved in pocket formation. Amino acid residues indicated by pink carbons are those whose main chains are involved in pocket formation. Amino acid residues labelled in red are those that have been added to pocket formation by the F208G mutation.

208th amino acid residue in the amino acid sequence of the AL domain and showed that the size of the residue's side chain could be a determinant of substrate specificity. Replacement with an amino acid residue with a smaller side chain comprising three or fewer atoms, which is preferable for producing C17 homologues, was predicted to extend the depth of the putative fatty acid-binding pocket. This suggests that the depth of the putative fatty acid-binding pocket is closely related to the substrate specificity of iturin synthetase. Our results are consistent with the structural basis of substrate specificity proposed for FAALs. Therefore, for other NRPSs with AL domains classified as FAAL[29], it may be possible to modify the fatty acid moiety of the product by modifying the substrate-binding pocket as in this study.

In strains with mutations in the AL domain, changes in the composition of lipopeptide homologues were accompanied by changes in the

productivity of lipopeptides (Figs. 2, 3a and Supplementary Figs. 4, 5). For example, the mycosubtilin productivity of BL063 harbouring MycAL was $1.1 \pm 0.1$ g/L, $68 \pm 1\%$ of which were C17 homologues. In contrast, the mycosubtilin productivity of BL123 harbouring ItuAL was $0.6 \pm 0.02$ g/L, $87 \pm 0.4\%$ of which were C15 homologues (Fig. 2a). Productivity is higher when the relative abundance of C17 homologues is higher in the lipopeptides produced by mutant strains harbouring the MycAL-type AL domain. One possible reason for this change in productivity is the correspondence between the substrate specificity of the AL domain and the fatty acid pool in the cell. Under our culture conditions, C15 fatty acids were the most abundant in cellular lipids (Supplementary Table 3). In iturin family lipopeptide synthetases, fatty acids are taken up by the AL domain and then elongated by two carbons by the adjacent KS domain (Fig. 1). This means that the substrates for the production of C15 and C17 homologues are C13

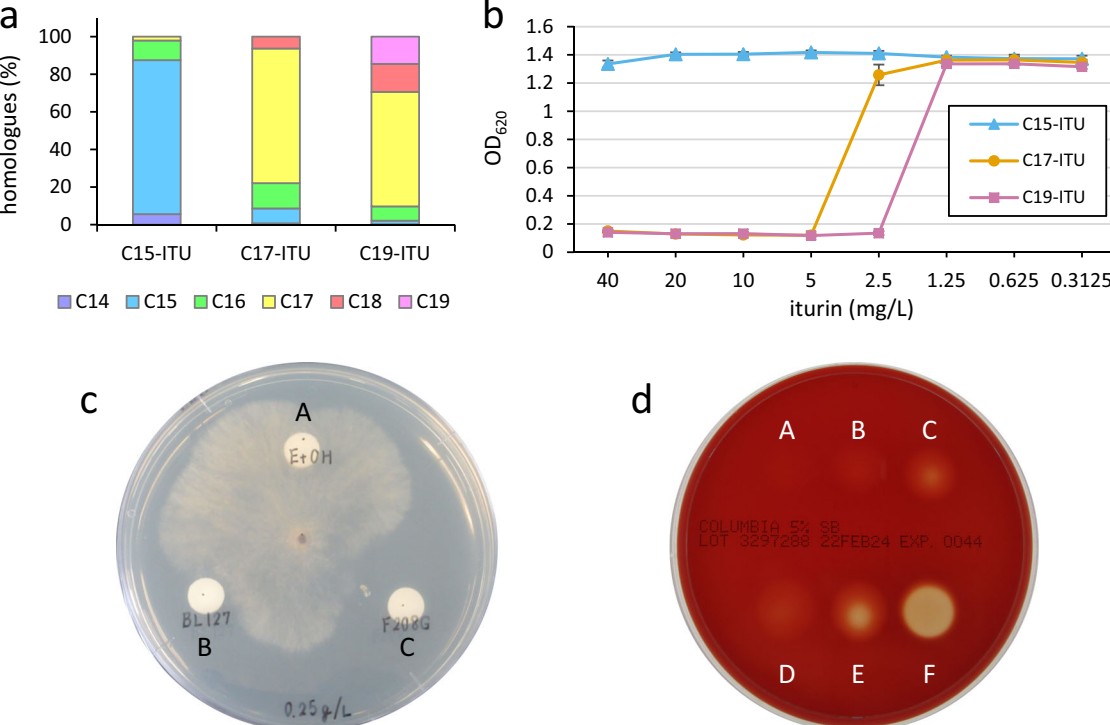

**Fig. 5 | Effect of fatty acid chain composition on iturin activity. a** Relative iturin homologue abundance in the iturin extracts. **b** In vitro dose-response curves obtained with iturin extracts (C15-ITU: blue triangles, C17-ITU: orange circles, C19-ITU: pink squares) using *Saccharomyces cerevisiae* BY4742. **c** Antifungal assays of iturin extracts against *Rhizoctonia solani*. A = ethanol (control), B = 250 mg/L

C15-ITU, C = 250 mg/L C19-ITU. **d** Haemolytic assay of iturin extracts using Columbia agar with 5% sheep blood. A = 500 mg/L C15-ITU, B = 1000 mg/L C15-ITU, C = 2000 mg/L C15-ITU, D = 500 mg/L C19-ITU, E = 1000 mg/L C19-ITU, and F = 2000 mg/L C19-ITU.

and C15 fatty acids, respectively. The fatty acid composition in *B. subtilis*, where C15 fatty acid is rich, is preferable for the production of C17 lipopeptide homologues. In addition to the favourable fatty acid composition for the production of C17 homologs, the substrate specificity of the MycAL-type AL domain is preferred for C15 fatty acids (Fig. 2). In strains harbouring the MycAL-type AL domain with a high C17 lipopeptide content, more C15 fatty acids should be consumed as substrates. In contrast, in strains harbouring the ItuAL-type AL domain with a high C15 lipopeptide content, more C13 fatty acids should be consumed as substrates. This preference for fatty acid chain length reflects the substrate specificity of the AL domain. The substrate specificity of the MycAL-type AL domain, which prefers C15 fatty acids, makes it possible to utilise abundant C15 fatty acid in the cell efficiently, resulting in high productivity. It may be possible to further improve productivity by efficiently supplying fatty acids with chain lengths corresponding to the fatty acid moiety of the lipopeptide produced as a substrate, for example, by genetically modifying the fatty acid supply pathway or by adding fatty acids to the culture medium.

Our results show that the longer the fatty acid moiety, the higher the antimicrobial activity. Various studies have reported the use of iturin family lipopeptides as antimicrobial agents[30–33]. In addition, the global trend to reduce chemical fungicide use has attracted attention to iturin family lipopeptides as alternative biofungicides. Iturin is mainly composed of C14–C16 homologues[18]. In the present study, we produced the iturin homologues with the fatty acid moiety of C18 and C19, which to our knowledge, have not been reported to date. We aimed to elucidate the terminal structures of their acyl chains; however, we were unable to purify these novel iturins. Therefore, further analysis is needed to elucidate the terminal structure. Compositions containing the newly obtained C19 iturin are promising candidates for new antimicrobial materials because of their high antimicrobial activity. If iturins with fatty acid moieties of C20 or longer can be produced by further modification of the AL domain, it may be possible to obtain compositions with higher antimicrobial activity. As an

example of modification of the AL domain, saturation mutations at the 287th and 332nd amino acid residues in the amino acid sequence of the AL domain, which are involved in determining the length of the fatty acid moiety, can be performed. However, as mentioned above, for efficient production, it may be necessary to supply fatty acids corresponding to the chain length of the fatty acid moiety of the lipopeptide as substrates. Therefore, a combination of modifying the fatty acid supply pathway and/or adding fatty acids to the culture medium must also be considered.

In this study, we demonstrated that the substrate specificity of the AL domain determines fatty acid moiety length in iturin family lipopeptides. Amino acid substitutions enlarging the substrate-binding pocket enhanced the specificity of the AL domain for longer fatty acids. This mechanism could be based on the universal mechanisms of FAALs. These findings provide a mechanism for rationally altering the lengths of fatty acid moieties in lipopeptides synthesised by NRPS. The amino acid moiety of lipopeptides has been examined for change by domain substitution or mutation[34–36]. Recently, there have been advances in research on efficient methods for domain substitution[37–39]. By combining these techniques, it may be possible in the future to freely customise lipopeptides according to their application.

## Methods

### Strains, plasmids, and growth conditions

The strains and plasmids used in this study are listed in Table 1. *B. subtilis* and *Escherichia coli* strains were cultivated at 37 °C in Luria–Bertani medium (LB). Appropriate antibiotics [100 µg/mL ampicillin, 5 (for *B. subtilis*) or 50 (for *E. coli*) µg/mL kanamycin, 5 µg/mL chloramphenicol, 100 µg/mL spectinomycin, or 0.5 µg/mL erythromycin] were used where necessary.

The yeast strain was cultured at 30 °C in YPD medium [10 g/L yeast extract, 20 g/L HIPOLYPEPTON (Nihon Pharmaceutical Co., Ltd., Osaka, Japan), and glucose 10 g/L]. The fungal strain was propagated at 25 °C on Difco Potato Dextrose Agar (PDA; BD Biosciences, San Jose, CA, USA).

For lipopeptide production, strains were inoculated into LB medium and then incubated overnight at 37 °C with shaking at 300 rpm as seed culture. Thereafter, 12.5 μL of the seed culture was inoculated into 2.5 mL of a production medium [40 g/L soybean flour (Sigma-Aldrich, St. Louis, MO, USA), 5 g/L K$_2$HPO$_4$, 0.5 g/L MgSO$_4$·7H$_2$O, 0.18 g/L CaCl$_2$·2H$_2$O, 0.025 g/L FeSO$_4$·7H$_2$O, and 0.022 g/L MnCl$_2$·4H$_2$O] in a test tube (φ25 mm × 200 mm) and cultured at 30 °C with shaking at 300 rpm for 72 h.

### Construction of mycosubtilin-producing mutant strains

The primers used in this study are listed in Supplementary Data 1. Bamba et al. constructed a mycosubtilin-producing strain, KB04[40], harbouring the mycosubtilin operon, which encodes mycosubtilin synthetase, from ATCC6633 and lpa-14[41], which encodes 4-phosphopantetheinyl transferase, from RB14. To enhance mycosubtilin production by the KB04, the *rplGB* promoter region was inserted upstream of the mycosubtilin operon. The upstream homologous arm of the mycosubtilin operon from ATCC6633, an erythromycin resistance cassette amplified from pMUTinHis[42], the promoter region of *rplGB* from *B. subtilis* 168, and the downstream homologous arm of the mycosubtilin operon promoter from ATCC6633 were amplified using the corresponding primers and fused using overlap extension polymerase chain reaction (OE-PCR). The KB04 was transformed using the fused fragment, and transformants were selected on LB agar plates containing 1 μg/mL erythromycin and 12.5 μg/mL lincomycin. The resulting mutant strain was designated BL043.

Subsequently, a markerless *sigF* deletion mutant strain of BL043 was constructed using the counter-selection marker *pheS*\* cassette[43]. The up- and downstream homologous arms of *sigF* from *B. subtilis* 168 and the spectinomycin resistance cassette from pJL62[44] were amplified using the corresponding primers. The *pheS*\* cassette was synthesised according to a previous report[43] using GENEWIZ (Azenta Life Sciences, Burlington, MA, USA). Here, we used the *pheS* sequence derived from *Bacillus amyloliquefaciens* DSM 7 as the *pheS*\* cassette, as mentioned in the report, to prevent undesirable homologous recombination between *pheS* on the *B. subtilis* 168 genome and *pheS*\* on the cassette. These fragments were fused using OE-PCR, after which they were transformed into BL043. Thereafter, the transformants were selected on LB agar plates containing 150 μg/mL spectinomycin. The resulting mutant strain was transformed using a fused fragment wherein the up- and downstream homologous arms of *sigF* from *B. subtilis* 168 were fused using OE-PCR. Subsequently, the transformants were selected on LB agar plates containing 1 mM 4-chloro-DL-phenylalanine (4CP). The resulting mutant strain was designated BL063.

To replace the AL and ACP domains, strain construction was performed in two steps using the *pheS*\* cassette. First, BL082 was obtained by transforming BL063 with the OE-PCR-fused fragment comprising the upstream homologous arm of the sequence of *mycAL*, which encodes the AL domain in mycosubtilin synthetase, from ATCC6633, *pheS*\* cassette, the spectinomycin resistance cassette from pJL62[44], and the downstream homologous arm of the sequence of *mycACP*, which encodes the ACP domain in mycosubtilin synthetase, from ATCC6633. Afterwards, the transformants were selected on LB agar plates containing 150 μg/mL spectinomycin. Second, BL082 was transformed with a fused fragment containing the upstream homologous arm of the sequence of *mycAL* from ATCC6633, the sequence encoding the AL domain, the sequence encoding the ACP domain, and the downstream homologous arm of *mycACP* from ATCC6633. Subsequently, the transformants were selected on LB agar plates containing 1 mM 4CP. BL123, BL134, BL135, BL136, BL137, BL138, and BL139 mutant strains were generated using the methods described above.

To identify the amino acid residues mediating the specificity of fatty acids as substrates, we generated amino acid substitution mutant strains using BL083 as a host strain and donor DNA constructed using the QuikChange® method (Agilent Technologies, Santa Clara, CA, USA). First, BL083 was obtained by transforming BL063 with the OE-PCR-fused fragment comprising the upstream homologous arm of the mycosubtilin operon from ATCC6633, the spectinomycin resistance cassette from pJL62[44], and the downstream homologous arm of the sequence of *mycACP* from ATCC6633. Afterwards, the transformants were selected on LB agar plates containing 150 μg/mL spectinomycin. Second, we constructed a pUC19-based plasmid (pUC19-mycU-erm-PrplGB-fenF-ituAL-mycACP-mycAD) using an In-Fusion® HD Cloning Kit (New England Biolabs, Ipswich, MA, USA) and *E. coli* JM109 (Takara Bio Inc., Shiga, Japan). The inserted fragment was an OE-PCR-fused fragment comprising the upstream homologous arm of the mycosubtilin operon from ATCC6633, the erythromycin resistance cassette from pMUTinHis[42], the promoter region of *rplGB* from *B. subtilis* 168, *fenF* and the intergenic region between *fenF* and *mycA* from ATCC6633, the sequence of *ituAL*, which encodes the AL domain in iturin synthetase, from RB14, the sequence of *mycACP*, and the downstream homologous arm of *mycACP* from ATCC6633. Third, we generated mutated plasmids using the QuikChange method with the primers listed in Supplementary Data 1, and then the QuikChange products were used to transform *E. coli* JM109. The resulting mutated plasmids were purified, linearised using *Sma*I, and transformed into BL083 cells. Thereafter, the transformants were selected on LB agar plates containing 1 μg/mL erythromycin and 12.5 μg/mL lincomycin. The resulting strain was designated XyyyZ, where X indicates the amino acid residue before mutation, yyy indicates the amino acid position, and Z indicates the amino acid residue after mutation.

### Construction of iturin-producing mutant strains

We constructed an iturin-producing strain using BL063 as the host strain and the iturin operon from RB14, which encodes iturin synthetase[45], as the donor DNA. First, BL063 cells were transformed using an OE-PCR-fused fragment comprising the up- and downstream homologous arms of the mycosubtilin operon from BL063 and the spectinomycin resistance cassette from pJL62[44]. Subsequently, the transformants were selected on LB agar plates containing 150 μg/mL spectinomycin. The resulting mycosubtilin operon-deficient mutant strain, BL113, was used as the host strain for insertion of the iturin operon. As the iturin operon is large, we divided it into six fragments, which we inserted in a stepwise manner. Before the final step of iturin operon insertion, *srfAB* disruption was performed to obtain an iturin monoproducer. To disrupt *srfAB*, a kanamycin resistance cassette from pAPNCK[46] was inserted downstream of *comS*, and the transformants were selected on LB agar plates containing 7.5 μg/mL kanamycin. Finally, we transformed the *srfAB*-deficient mutant strain BL144 with a pUC19-based plasmid (pUC19-mycU-erm-PrplGB-ituD-ituAL-ituACP-ituAD) containing an OE-PCR-fused fragment comprising the upstream homologous arm of the mycosubtilin operon from ATCC6633, the erythromycin resistance cassette from pMUTinHis[42], the promoter region of *rplGB* from *B. subtilis* 168, *ituD* and the intergenic region between *ituD* and *ituA* from RB14, the sequence of *ituAL*, the sequence of *ituACP*, which encodes the ACP domain in iturin synthetase, and the downstream homologous arm of *ituACP* from RB14. Before transformation, the plasmid was purified and linearised using *Xho*I. BL127, F208L, K287R, A332T, double mutant (F208L and K287R), triple mutant (F208L, K287R, and A332T) strains were generated according to the method described above. To obtain BL128, we transformed BL144 with a pUC19-based plasmid (pUC19-mycU-erm-PrplGB-ituD-mycAL-ituACP-ituAD) containing an OE-PCR-fused fragment comprising the upstream homologous arm of the mycosubtilin operon from ATCC6633, the erythromycin resistance cassette from pMUTinHis[42], the promoter region of *rplGB* from *B. subtilis* 168, *ituD* and the intergenic region between *ituD* and *ituA* from RB14, the sequence of *mycAL* from ATCC 6633, the sequence of *ituACP* and the downstream homologous arm of *ituACP* from RB14. Before transformation, the plasmid was purified and linearised using *Xho*I.

To generate strains with ItuAL in which the 208th Phe residue was replaced by proteinogenic amino acids other than Phe, we used donor DNA constructed using the QuikChange® method described above and BL144 as the host strain. First, we generated mutated plasmids using the QuikChange

method with pUC19-mycU-erm-PrplGB-ituD-ituAL-ituACP-ituAD and the primers listed in Supplementary Data 1. Second, the resulting mutated plasmids were purified, linearised using *Xho*I, and transformed into BL144 cells. Subsequently, the transformants were selected on LB agar plates containing 1 μg/mL erythromycin and 12.5 μg/mL lincomycin. The resulting strain was designated F208X, where X indicates the amino acid residue after the mutation.

To generate strains with MycAL in which the 208th Leu residue was replaced by other amino acids, we used donor DNA constructed using the QuikChange® method described above and BL144 as the host strain. We generated mutated plasmids using the QuikChange method with pUC19-mycU-erm-PrplGB-ituD-mycAL-ituACP-ituAD and the primers listed in Supplementary Data 1. The resulting strain was designated MycAL_L208X, where X indicates the amino acid residue after the mutation.

## Lipopeptide analysis

To determine the lipopeptide productivity of the various strains, the lipopeptides contained in the culture broth were extracted using methanol. Samples were analysed by high-performance liquid chromatography (HPLC) using an HPLC Prominence (Shimadzu Corporation, Kyoto, Japan) equipped with an NPS ODS-IIIE column (4.6 mm × 33 mm, particle size 1.5 μm; Promigen Life Sciences, LLC, Downers Grove, IL, USA) at a temperature of 40 °C. A gradient of acidified water (0.1% formic acid) (solvent A) and acidified acetonitrile (0.1% formic acid) (solvent B) was used as the mobile phase at a constant flow rate of 0.7 mL/min, starting at 10% B and increasing to 51% B in 5 min. Solvent B was maintained at 51% for 15 min before being returned to its initial concentration. The injection volume was 10 μL, and the detection wavelength was 205 nm. The concentration of the lipopeptides was calculated using iturin A (CAS 52229-90-0, purity 95%; Sigma-Aldrich) as a standard. The relative abundance of the lipopeptide homologues was calculated from the area of each peak as the sum of the area values of the peaks corresponding to each homologue.

For liquid chromatography time-of-flight mass spectrometry (LC-TOF-MS) analysis, a sample volume of 10 μL was injected at 0.7 mL/min and analysed by HPLC using a Nexera X2 UHPLC System (Shimadzu Corporation) equipped with a Kinetex EVO C18 column (4.6 mm × 150 mm, particle size 2.6 μm; Phenomenex, Torrance, CA, USA) at a temperature of 40 °C. A mixture of water, acetonitrile, and formic acid (60:40:1, v/v/v) was used as the mobile phase. Mass spectra were obtained using a TripleTOF 6600+ System (SCIEX, Framingham, MA, USA) equipped with a standard ESI source. The collision energy was set to 60 V. The first quadrupole mass spectrometer selected and fragmented the hydrogen-ionised molecules of interest. The resulting hydrogen-ionised fragments were analysed and recorded using time-of-flight secondary ion mass spectrometry.

## Expression and purification of the recombinant protein

We obtained pET28a-ituAL-ACP1 and pET28a-ituAL_F208G-ACP1 using pET28a (Merck, Darmstadt, Germany) as the parent plasmid and *E. coli* BL21-AI (Thermo Fisher Scientific, Waltham, MA, USA) was transformed using this plasmid. The transformants were inoculated into LB medium containing 50 μg/mL kanamycin and incubated at 37 °C for 16 h. Overnight cultures were transferred into LB medium containing kanamycin to adjust the $OD_{600}$ to 0.1 and incubated at 37 °C until the $OD_{600}$ reached 0.4. Thereafter, L-arabinose and IPTG were added at final concentrations of 0.2% and 0.1 mM, respectively, and expression was induced at 20 °C for 16 h. The collected pellet was resuspended in lysis buffer (500 mM NaCl, 20 mM HEPES (pH 8.0), 5 mM imidazole, 5 mM $MgCl_2$, 10% glycerol, 10% Sorbitol, 1 mM DTT), after which cOmplete™, EDTA-free protease inhibitor cocktail (Roche, Basel, Switzerland), and lysozyme were added. Subsequently, the cells were lysed in an ice bath using an ultrasonic homogeniser (VIOLAMO; AXEL GLOBAL, Osaka, Japan). The soluble fraction was collected by centrifugation at 12,000 × *g* for 20 min, whereas the insoluble fraction was completely removed by passing it through an ultrafiltration filter (pore size 0.22 μm). Next, 1 mL of the cOmplete His-tag purification

column (Roche) was mounted on an AKTA start purification system (Cytiva, Marlborough, MA, USA) and equilibrated using a lysis buffer. Afterwards, the target protein was subjected to gradient elution using elution buffer [500 mM NaCl, 20 mM HEPES (pH 7.5), 500 mM imidazole, 5 mM $MgCl_2$, 10% glycerol, 40% sorbitol, and 1 mM DTT]. The eluted fraction was concentrated with Amicon Ultra 0.5 K (Merck) and replaced elution buffer with storage buffer [300 mM NaCl, 20 mM HEPES (pH 7.5), 5 mM $MgCl_2$, 40% sorbitol]. The protein concentration was determined using the Bradford method.

## Fatty acid preparation

Fatty acids *n*-C10, *n*-C11, *n*-C12, *n*-C13, *n*-C14, *n*-C15, *anteiso*-C15, *iso*-C15, and *n*-C16 were purchased from Tokyo Chemical Industry Co., Ltd. (Tokyo, Japan) or Avanti Polar Lipids, Inc. (Alabaster, AL, USA). Each fatty acid was dissolved in DMSO to a concentration of 100 mM and used as a stock solution. Next, the fatty acids were diluted to 5 mM using 0.05% Brij35 prepared in 50% DMSO. Subsequently, a 50% DMSO solution containing 5 mM of each fatty acid and 0.05% Brij35 was used in the reaction as a 5 × fatty acid solution. Warming was continued at 45 °C until just before the reaction to completely dissolve the fatty acids.

## Measurement of AL activity

A 5× reaction buffer [250 mM HEPES (pH 7.5), 250 mM $MgCl_2$, 50% sorbitol, and 20 mM tris(2-carboxyethyl)phosphine hydrochloride] was prepared and passed through an ultrafiltration filter (pore size 0.22 μm). The reaction mixture [ituAL-ACP1 or ituAL_F208G-ACP1; 10 μM, fatty acid; 1 mM, Brij35; 0.01%, ATP; 1 mM, HEPES (pH 7.5); 50 mM, $MgCl_2$; 50 mM, sorbitol; 10%, DMSO; 30%] at a 75 μL reaction volume was incubated at 30 °C for 30 min. The reaction was stopped by addition of a 25 μL stop solution to a final concentration of 100 mM EDTA (pH 8.0) and 20% acetonitrile. A reaction with no enzyme added was used as a negative control. The centrifuged supernatants were analysed using HPLC.

## Analysis of enzymatic activity using HPLC

HPLC analysis was performed using an Alliance 2796 System (Waters Corporation, Milford, MA, USA) and a UV/Vis detector 2489 (Waters Corporation). Separation was performed isocratically using a weak anion exchange column Axpak WA-624 (6.0 mm × 150 mm, particle size 10 μm; Shodex, München, Germany) as the separation column and 0.4 M $NaH_2PO_4$: acetonitrile = 4:2 as the mobile phase at a flow rate of 1.2 mL/min. The detector was set at 254 nm and the column temperature at 50 °C. The injection volume was 20 μL.

## Structure prediction of AL domain

ColabFold v1.5.5[47] was used to predict the structures of the native and F208G variants of the AL domain of iturin A synthetase. The amino acid sequence of native ItuAL with the mycosubtilin synthetase ACP domain is shown in Supplementary Fig. 9. For both predictions, the template, MSA, and pair modes were "PDB100", "mmseqs2_uniref_env", and "unpaired_paired", respectively. The top-ranked structures in each prediction were subjected to structural relaxation using Amber. For the native and F208G variant-predicted structures, the average local distance difference test (lDDT) values over 444 amino acid residues of the sub-domains in which the fatty acid recognition pocket resides (coloured in yellow and blue in Supplementary Fig. 8) were 94.3 and 94.0, respectively. The lDDT of all amino acid residues contributing to pocket formation were >90 in both predictions. In the same way, the structural predictions of the native and L208G of mycosubtilin synthetase were performed using the amino acid sequence shown in Supplementary Fig. 2.

## Selection of amino acid residues forming the substrate-binding pocket in the predicted structure

Initially, the position of the substrate-binding pocket was detected using cavenv in CCP4[48]. The fatty acid model with C12 for the native prediction model or with C15 for the F208G model was fitted to the pocket using the

coot programme in CCP4[48]. The position of the carboxyl group of fatty acids was determined based on the position of the acyl group of phenylalanyl adenylate in the crystal structure of MycG A-PCP (PDB ID: 4R0M)[49] superimposed on the predicted model. Superimposition of MycG A-ACP onto the predicted ItuAL model was performed using the secondary structure matching alignment mode of coot in CCP4[48]. In this structural alignment, 523 amino acid residues were aligned with 1.92 Å of the root mean square deviation. The amino acid residues that exist at a 5 Å distance from the fatty acid model were selected as candidates for the amino acid residues comprising the substrate-binding pocket. Each amino acid residue was visually evaluated to determine whether it contributed to pocket surface formation.

### Lipopeptide extraction

For lipopeptide extraction, strains were precultured in LB medium and then incubated overnight at 37 °C with shaking at 300 rpm as seed cultures. Next, 250 μL of the seed culture was inoculated into 50 mL of a production medium in a 500 mL shaking flask and cultured at 30 °C with shaking at 130 rpm for 72 h. Thereafter, the culture broth was centrifuged at $9100 \times g$ for 5 min, after which the pellet was resuspended in ethanol. The obtained suspension was centrifuged at $7300 \times g$ for 5 min, and then the supernatant was filtrated using a 0.2-μm filter (13HP020AN; Advantec Toyo Kaisha, Ltd., Tokyo, Japan). The obtained filtrate was dried and dissolved in ethanol, whereas the obtained suspension was centrifuged at $20,400 \times g$ for 5 min. Finally, the supernatant was analysed by HPLC, as described in the section "Lipopeptide analysis", and used for further analysis.

### Determination of minimal inhibitory concentrations

*S. cerevisiae* BY4742 was inoculated into a test tube ($\phi 25 \text{ mm} \times 200 \text{ mm}$) containing 1 mL of YPD medium and subjected to incubation for 19 h at 30 °C with shaking at 300 rpm as a G1 seed culture. Next, the G1 seed culture was transferred to a test tube containing 3 mL of YPD medium to adjust the $OD_{600}$ to 0.1 and incubated for 5 h at 30 °C with shaking at 300 rpm as a G2 seed culture. Afterwards, the G2 seed culture was diluted in YPD medium to adjust the $OD_{600}$ to 0.01, and 490 μL of the obtained suspension was transferred into each well of a 96-well deep-well plate (BM6030S; BM Equipment Co., Ltd., Tokyo, Japan). Next, 10 μL of lipopeptide ethanol extract (2000, 1000, 500, 250, 125, 62.5, 31.25, or 15.625 mg/L) was added to each well and mixed thoroughly. The deep-well plate was sealed using AirPore Tape Sheets (Qiagen, Hilden, Germany) and then incubated for 24 h at 30 °C with shaking at 1000 rpm. Fungal growth was measured at 620 nm using a plate reader (MULTISKAN FC; Thermo Fisher Scientific).

### Analysis of antifungal activity

Paper discs (49005010; Advantec Toyo Kaisha, Ltd.) were placed on the PDA plates, and then 50 μL of ethanol (control) or 250 mg/L lipopeptide ethanol extract was spotted on the paper disc. Subsequently, a 2 mm fungal plug of *R. solani* was placed in the centre of the PDA plate, after which the plates were incubated at 25 °C for 4 days.

### Evaluation of haemolytic activity

Ten microlitres of 500, 1000, or 2000 mg/L lipopeptide ethanol extract were spotted on BD BBL™ Columbia agar with 5% sheep blood (BD Biosciences), which was then incubated at 25 °C. Haemolytic activity was assessed after 7 days based on the development of a clear zone.

### Lipid analysis

To determine the relative abundance of each fatty acid in cells, total lipid extraction was performed using a modification of the method of Bligh and Dyer according to a previous report[21]. The culture broth of the F208G mutant strain prepared as described in the section "Lipopeptide extraction" was centrifuged at $8000 \times g$ for 5 min, after which the pellet was resuspended in 35 mL 1% (w/v) NaCl. The washed pellet was frozen at $-80$ °C, and then lyophilised. The lyophilised pellet was crushed and used for total lipid

extraction. To 250 mg of the lyophilised pellet, 25 mL of chloroform, 50 mL of methanol, and 20 mL of water were added in this order and mixed thoroughly. After allowing to stand overnight at room temperature, 25 mL of chloroform and 25 mL of water were added and mixed well. The lower chloroform phase was collected and dried to obtain a total lipid sample. A production medium without inoculation of the strain was cultured under the same conditions and used as a control.

The total lipid samples were used to prepare the fatty acid methyl esters with a fatty acid methylation kit (NACALAI TESQUE, INC., Kyoto, Japan). Fatty acid methyl ester samples were purified with a fatty acid methyl ester purification kit (NACALAI TESQUE, INC.) and analysed through gas chromatography-mass spectrometry (GC-MS).

GC-MS analysis was performed according to a previous report[21]. An Agilent 8890 gas chromatograph with a 5977B mass-sensitive detector equipped with an Agilent HP-5ms Ultra Inert column (30 m × 0.25 mm i.d. × 0.25 μm film thickness) was used (Agilent Technologies). The following GC temperature program was applied: 60 °C (held for 2 min), 20 °C/min ramp to 170 °C, 5 °C/min ramp to 240 °C, 30 °C/min ramp to 320 °C (held at 320 °C for 2 min). Each peak was assigned based on *m/z* according to a previous report[21]. As an exception, the peak of C18 fatty acid was estimated from m/z and soybean oil composition because soybean flour was included in the production medium. The relative abundance of each fatty acid was calculated from the area of each peak as the sum of the area values of the peaks corresponding to each fatty acid.

### Reporting summary

Further information on research design is available in the Nature Portfolio Reporting Summary linked to this article.

## Data availability

The authors declare that the main data supporting the findings of this study are available within the article and its Supplementary Information. Alternatively, the data are available from the corresponding author upon reasonable request.

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

## Acknowledgements

We thank Dr Makoto Shoda from Institute of Science Tokyo and Dr Kenji Tsuge from Kobe University for their kindly providing genomic DNA of *B. subtilis* RB14. We also thank Yuko Funabashi and Taiki Saito from KANEKA TECHNO RESEARCH CORPORATION for their valuable help with LC-MS analysis, and Saya Kato and Tomoki Hayashi from KANEKA CORPORATION for their technical help with GC-MS analysis. This work was supported by research funding JPNP20011 from the New Energy and Industrial Technology Development Organization (NEDO).

## Author contributions

R.A., N.T., Y.O., and S.K. conceived and designed the study. R.A., E.K., K.K., and H.A. performed the experiments. R.A., E.K., K.K., and H.A. performed data analysis and prepared all figures and tables. H.A., N.S., T.H., N.T., Y.O., and S.K. supervised the research planning, experiments, and interpretation of results. R.A., E.K., and H.A. wrote the manuscript. R.A., E.K., H.A., N.S., Y.O., and S.K. edited the manuscript, and all authors approved the final version.

## Competing interests

The authors declare no competing interests.
