## [Peer review file · Communications Chemistry]

Engineering of acyl ligase domain in non-ribosomal peptide synthetases to change fatty acid moieties of lipopeptides

Corresponding Author: Dr Shingo Kobayashi

Version 0:

Reviewer comments:

Reviewer #1

(Remarks to the Author)

Engineering the length of the fatty acid chain of NRPS-derived lipopeptides is an interesting topic because it can balance the activity and toxicity of lipopeptides. In this manuscript, Aoki et al reported the engineering of acyl ligase domain in NRPS to change the fatty acid chain, which is an indispensable supplement to the previous engineering the starter condensation of domain of NRPS for fatty acid chain design. This manuscript firstly verified the fatty acyl ligase domain is responsible for the length of the fatty acid moiety of iturin family lipopeptides. The subdomain or domain exchange, and point mutation experiments as well as the 3D structure prediction suggested the substrate-binding pocket of fatty acid chains and the key site to control the length, which can be beneficial for change the distribution of different derivatives in vivo by mutagenesis. This work shed lights on the future for design of fatty acyl moieties for lipopeptides.

Some concerns need to be addressed before recommendation for acceptance

For the title, "design" is not suitable for the manuscript, can be changed to "change"

For the key amino acid residue at position 208, F for iturin and L for mycosubtilin, the larger sequence alignment or phylogenetic analysis of AL domains (not only show two of them in figure S1) is required to show the potential relationship between the atom number of side chain of amino acid 208 and the length of fatty acyl moiety. This analysis result can be used to accurately predict the structure of lipopeptide for genome mining of novel peptides.

Figure 1 the stereochemistry of iturin should be present.

Line 77 ituD should be italic

The exact structure of two C18- and two C19-iturins should be further elucidated by NMR, the MS/MS cannot distinguish these isomers.

Reviewer #2

(Remarks to the Author)

This was a very comprehensive study looking into the source of fatty acid specificity in lipopeptide antibiotic biosynthesis. The amount of work contained in this paper is significant and the work seems to have been very well performed. I have some minor comments that the authors may wish to consider in a revision:

The introduction is very "light on" with regards NRPS introductory references. There are a lot of structures discussed in the introduction and these should be included in a figure to help the reader. Also, discussion of C-domains should be more extensive; also the way E and C domains work together.

There are a few minor language issues, like line 84, where it should be: "an asparagine", line 87 "stepwisely" etc

Discussion of different acyl chain lengths - discuss iso/ anteiso branched species (are any present?) I know these are mentioned later but not early on.

Regarding errors measurements discussed in line 143 - can't be +/- 0.0, shouldn't it be 0.0X (2 significant figures)...?

Line 150 - regarding the division of enzymes into regions, how was this chosen? Secondary structure predictions?

When looking into the different specificities of the chimeric constructs, why didn't the authors try to incorporate specific subsections into one enzyme? That would have helped to see if the changes in specificity were localised into such a region. This also speaks again to a need to better explain how these sections were selected and how the link to the predicted active site/ structure of the domains.

For the mutants, how were the amino acids chosen?

I appreciate that the mutations were made prior to structural production, but I would like to see a bit more integration of mutation data into discussion of the model besides residue 208.

Figure 4 - how are the structures predicted (mention in the caption)? I think a different representation should be shown for the predicted binding pocket as the way it is shown currently looks like omit density from a crystal structure (which it is not).

The activity data for these compounds is very well performed and interesting.

The discussion was well written and interesting - I would have loved to find more details out about the relative composition of different fatty acid species in the cell, as this important for the study.

Reviewer #3

(Remarks to the Author)

In this manuscript, Aoki et.al. have attempted to modulate the fatty acid specificity of an NRPS module responsible for the production of iturin family of cyclic lipopeptides (CLPs). How different family of CLPs show different fatty acyl chain profiles is not well understood. Through various mutational analysis, the authors identify fatty acyl-AMP ligase (ALs) as the key factor determining the acyl chain length of these molecules and demonstrate that it is possible to generate iturins with different acyl chains by mutating the ALs. They have further characterised the antifungal and haemolytic activities of these iturin variants and identified that long acyl chain-containing iturins exhibit better activity. The study has important implications in the generation of novel antifungal variants.

Some Concerns:

1. What is the rationale behind selecting residues for mutational analysis? This can be explained better in the revised manuscript.
2. Why only F208 was selected for saturation mutagenesis?
3. What is the reason for performing experiments with chimeric AL domains? Since the identified FAAL domain has been shown to be responsible for the specificity, a mutational analysis of the acyl chain binding pocket could have done the job. The authors may want explain why they adopted such a strategy!
4. Why was the in vitro activity of ALs on C-17 and C-19 fatty acids not tested for the F208G mutant, despite their higher antifungal activity?

Version 1:

Reviewer comments:

Reviewer #1

(Remarks to the Author)

Thanks for the revisions. I am basically satisfied with the revisions.

Reviewer #2

(Remarks to the Author)

The authors have made a very detailed and comprehensive effort to address the comments and suggestions of the reviewers. I am satisfied that the revised manuscript is now suitable for publication.

Reviewer #3

(Remarks to the Author)

The authors have addressed my concerns satisfactorily.

RESPONSE TO REVIEWERS

Dear Publishing Editor, Dear Reviewers,

First of all, we would like to express our sincere gratitude to all of you for having accepted to review this article, for your time, for your effort, and for all the constructive comments that you have made on this manuscript.

According to these comments, we have intensively revised our manuscript in order to make it clearer, more efficient and we did our best to address the points raised by the reviewers, in a point-by-point response. Our responses to the reviewers' comments, along with the corresponding modification added in the manuscript, are written in **red color**.

Response to the comments from Reviewer 1

Engineering the length of the fatty acid chain of NRPS-derived lipopeptides is an interesting topic because it can balance the activity and toxicity of lipopeptides. In this manuscript, Aoki et al reported the engineering of acyl ligase domain in NRPS to change the fatty acid chain, which is an indispensable supplement to the previous engineering the starter condensation of domain of NRPS for fatty acid chain design. This manuscript firstly verified the fatty acyl ligase domain is responsible for the length of the fatty acid moiety of iturin family lipopeptides. The subdomain or domain exchange, and point mutation experiments as well as the 3D structure prediction suggested the substrate-binding pocket of fatty acid chains and the key site to control the length, which can be beneficial for change the distribution of different derivatives in vivo by mutagenesis. This work shed lights on the future for design of fatty acyl moieties for lipopeptides.

Som concerns need to be addressed before recommendation for acceptance.

[Comment 1]

For the title, "design" is not suitable for the manuscript, can be changed to "change".

>[Response]

Thank you for your suggestion. I changed the title as follows;

"Engineering of acyl ligase domain in NRPS to **change** fatty acid moieties of lipopeptides".

[Comment 2]

For the key amino acid residue at position 208, F for iturin and L for mycosubtilin, the larger sequence alignment or phylogenetic analysis of AL domains (not only show two of them in figure S1) is required to show the potential relationship between the atom number of side chain of amino acid 208 and the length of fatty acyl moiety. This analysis result can be used to accurately predict the structure of lipopeptide for

genome mining of novel peptides.

>[Response]

I really appreciate your valuable suggestion. I added the sequence alignment as Supplementary Fig. 6 in accordance with your suggestion. F208 is highly conserved in iturin family lipopeptide synthetases, however the corresponding amino acid is exceptionally leucine in the mycosubtilin synthetase derived from *Bacillus subtilis* ATCC6633.

Based on the above, the following text was added in lines 216-218.

“The F208 residue is highly conserved in the AL domain of iturin family lipopeptide synthetases, except for the mycosubtilin synthetase derived from ATCC6633 in which the corresponding amino acid residue is Leu (Supplementary Fig. 6).”

[Comment 3]

Figure 1 the stereochemistry of iturin should be present.

>[Response]

Thank you for your suggestion. I corrected Figure 1 according to your suggestion.

[Comment 4]

Line 77 ituD should be italic.

>[Response]

Thank you for pointing out. I corrected the mistake.

[Comment 5]

The exact structure of two C18- and two C19-iturins should be further elucidated by NMR, the MS/MS cannot distinguish these isomers.

>[Response]

I really appreciate your valuable suggestion. To elucidate the end structure of fatty acids in C18- and C19-iturins, we tried hydrophobic interaction chromatography for purification. However, that was quite difficult maybe due to the soybean flour in the culture medium. I recognize that the elucidation of the end structure is very important for our research, but I consider that our results are sufficient to support the main points of this study, that is the underlying mechanism for decision of fatty acid length in iturin synthesizing process.

Based on the above, the following text was added in lines 430-434.

“In this study, we produced the iturin homologues with the fatty acid moiety of C18 and C19, which have

not been reported to date. We tried to elucidate the terminal structures of their acyl chains, however the purification of these novel iturins did not work. Therefore, the elucidation of the terminal structure requires further analysis.”

Response to the comments from Reviewer 2

This was a very comprehensive study looking into the source of fatty acid specificity in lipopeptide antibiotic biosynthesis. The amount of work contained in this paper is significant and the work seems to have been very well performed. I have some minor comments that the authors may wish to consider in a revision:

[Comment 1]

The introduction is very “light on” with regards NRPS introductory references. There are a lot of structures discussed in the introduction and these should be included in a figure to help the reader. Also, discussion of C-domains should be more extensive; also the way E and C domains work together.

>[Response]

I really appreciate your valuable suggestion. I added the chemical structure of CLPs as Supplementary Fig. 1 in accordance with your suggestion. Also, I added the explanation about the role of each domain in iturin synthetases.

Based on the above, the text was added in lines 57-83.

Notably the following text about the relationship between E and C domains was added in lines 70-76.

“The ^DC_L domain, the C domain downstream of the E domain, catalyses peptide bond formation between a D-amino acid and an L-amino acid. The E domain catalyses the epimerisation of an L-amino acid or the C-terminal L-amino acid of a growing peptide chain attached to the adjacent PCP domain into the D-configuration. Because the E domain provides a mixture of L- and D-amino acids, the ^DC_L domain is responsible for selecting the correct D-amino acid.”

[Comment 2]

There are a few minor language issues, like line 84, where it should be: “an asparagine”, line 87 “stepwisely” etc

>[Response]

Thank you for pointing out. I corrected the mistake.

[Comment 3]

Discussion of different acyl chain lengths - discuss iso/ anteiso branched species (are any present?) I know these are mentioned later but not early on.

>[Response]

I apologize for confusion. Not necessarily a lot, but some information about acyl chain length and iso / anteiso branched structure is mentioned in lines 51-53. The detail information about the relationship between length and species, for example C16 acyl chain mainly has n- / iso structure and C17 has iso / anteiso structure, is not mentioned. I recognize that this information is important, but a little too detailed as introduction.

Based on the above, the following text was added in lines 276-282.

“In LC-TOF-MS analysis of iturin produced by the F208G mutant strain (Supplementary Table 1), a peak corresponding to the C14 homologue and two peaks each corresponding to C15, C16, C17, C18, and C19 homologues were detected. These two peaks were assumed to correspond to homologues with n- and iso-fatty acid moieties for homologues with even-numbered fatty acid moieties, and to homologues with iso- and anteiso-fatty acids moieties for homologues with odd-numbered fatty acid moieties, respectively.”

[Comment 4]

Regarding errors measurements discussed in line 143 - can't be +/- 0.0, shouldn't it be 0.0X (2 significant figures)...?

[Response]

Thank you for pointing out. I corrected the mistakes.

[Comment 5]

Line 150 - regarding the division of enzymes into regions, how was this chosen? Secondary structure predications?

>[Response]

Thank you for pointing out. We divided enzymes into regions only by lengths, not by secondary structure predictions. However, the division points were chosen in regions conserved between ItuAL and MycAL as shown in Supplementary Fig. 2.

To avoid misunderstandings, the following text was added in lines 173-179.

“The AL domain was divided into five regions, each containing approximately 100 amino acids (Supplementary Fig. 2), and four types of chimaeras were constructed (Fig. 2b, Table 1). The boundary of the region was set at the part where the amino acid sequence was identical in the two sequences to prevent

the decrease in activity of the chimaeric AL domains. Because multiple amino acids may be involved in chain length determination, we successively replaced the ituAL sequence with the MycAL sequence.”

[Comment 6]

When looking into the different specificities of the chimeric constructs, why didn't the authors try to incorporate specific subsections into one enzyme? That would have helped to see if the changes in specificity were localised into such a region. This also speaks again to a need to better explain how these sections were selected and how the link to the predicted active site/ structure of the domains.

>[Response]

I really appreciate your valuable suggestion. We employed chimeric strategy in this report because that is easier in strain construction than the incorporation of specific subsections into ituAL. Also, as mentioned in the later section, we tested the single amino acid substitutions toward all positions which is different between ituAL and mycAL, and finally revealed that F208L, K287R and A332T have effects on the acyl chain lengths of iturin, but others have not. There are no contradictions between the results of chimaeras and the results of amino acid substitutions. I recognize that the incorporation of subsections is helpful for readers, however I consider that our existing results are sufficient to support the main points of this study, that is F208L, K287R and A332T have effects on the acyl chain length of iturin.

[Comment 7]

For the mutants, how were the amino acids chosen?

>[Response]

Thank you for your question. As mentioned above, we tested the single amino acid substitutions against all positions which is different between ituAL and mycAL.

To help readers understand, I added the following text in lines 200-204.

“For example, the 111th amino acid of ItuAL is His, and the corresponding amino acid residue of MycAL is Asn (Supplementary Fig. 2). Therefore, we generated a strain containing a mutant ItuAL in which the 111th His was substituted by Asn and named it H111N. According to this rule, we generated 48 mutant strains containing ItuAL with an amino acid substitution mutation...”

[Comment 8]

I appreciate that the mutations were made prior to structural production, but I would like to see a bit more integration of mutation data into discussion of the model besides residue 208.

>[Response]

I really appreciate your valuable suggestion. As shown in Fig. 3, K278R and A332T affected on the length

of fatty acid moiety besides F208L. I discussed the relationship between these mutations and the structural model in the paragraph “Predicted structure of AL domain”. However, we did not perform additional experiments on K278 and A332 such as saturated mutagenesis, because the effects of K278R and A332T were much smaller than that of F208L. Therefore, it is difficult to discuss any more about the structural change by the mutations on K278 and A332 on the basis of existing results.

[Comment 9]

Figure 4 - how are the structures predicted (mention in the caption)? I think a different representation should be shown for the predicted binding pocket as the way it is shown currently looks like omit density from a crystal structure (which it is not).

>[Response]

I apologize for confusion regarding your first question. As mentioned in lines 314, 712 and 728, the structures were predicted mainly by ColabFold and cavenv. I am sorry to omit the detailed explanation in the figure legends, because these methods are too much to be described.

I really appreciate your secondary suggestion. I modified Fig. 4 and its explanations in accordance with your suggestion.

[Comment 10]

The activity data for these compounds is very well performed and interesting.

>[Response]

Thank you for your comment.

[Comment 11]

The discussion was well written and interesting - I would have loved to find more details out about the relative composition of different fatty acid species in the cell, as this important for the study.

>[Response]

I really appreciate your valuable suggestion. We performed the additional experiment about fatty acid compositions under our fermentation conditions and added the results as Supplementary Table 3. As shown, C16 and C18 fatty acids were detected in the production medium, which is probably due to the soybean flour. C15 and C17 fatty acids were detected in the culture of F208G, suggesting that our strains mainly have C15 ~ C17 fatty acids under our cultivating condition. On the other hand, fatty acids in iturins were affected by substituting AL domains or mutating on F208, supporting the possibility that these modifications affected the substrate specificities of AL domains.

Based on the above, the following text was added in lines 283-294.

“We performed lipid analysis of the F208G mutant strain cultured in a production medium (Supplementary Table 3). Since the production medium contains soybean flour, precipitates of the culture broth contain cells and soybean flour. Therefore, a production medium without inoculation of the strain was analysed in the same way. C15, C16, C17, and C18 fatty acids were detected in precipitates of the culture broth of the F208G mutant strain. However, only a small amount of C14 fatty acid was detected (0.2 ± 0.02 %), and fatty acids with chain length less than C13 were not detected. n-C16 and n-C18 fatty acids were detected in precipitates of a production medium without inoculation of the strain. Since n-C16 and n-C18 fatty acids may be derived from soybean flour, it was suggested that at least iso-C14, iso-C15, anteiso-C15, iso-C16, iso-C17, and anteiso-C17 fatty acids were derived from the F208G mutant. Among these fatty acids, anteiso-C15 was the most abundant (11.6 ± 0.6 %).”

Response to the comments from Reviewer 3

In this manuscript, Aoki et.al. have attempted to modulate the fatty acid specificity of an NRPS module responsible for the production of iturin family of cyclic lipopeptides (CLPs). How different family of CLPs show different fatty acyl chain profiles is not well understood. Through various mutational analysis, the authors identify fatty acyl-AMP ligase (ALs) as the key factor determining the acyl chain length of these molecules and demonstrate that it is possible to generate iturins with different acyl chains by mutating the ALs. They have further characterised the antifungal and haemolytic activities of these iturin variants and identified that long acyl chain-containing iturins exhibit better activity. The study has important implications in the generation of novel antifungal variants.

[Comment 1]

What is the rationale behind selecting residues for mutational analysis? This can be explained better in the revised manuscript.

>[Response]

I really appreciate your valuable suggestion. we tested the single amino acid substitutions toward all positions which is different between ituAL and mycAL. There is no rationale behind selecting residues, such as structural predictions.

The following text was added in lines 200-204, in accordance with your suggestion.

“For example, the 111th amino acid of ItuAL is His, and the corresponding amino acid residue of MycAL is Asn (Supplementary Fig. 2). Therefore, we generated a strain containing a mutant ItuAL in which the 111th His was substituted by Asn and named it H111N. According to this rule, we generated 48 mutant strains containing ItuAL with an amino acid substitution mutation...”

[Comment 2]

Why only F208 was selected for saturation mutagenesis?

>[Response]

Thank you for your question. We revealed that F208L, K287R and A332T have effects on the acyl chain lengths of iturin, but others have not. As mentioned in lines 207-208, F208L had the biggest effect among these mutations, hence we performed the saturation mutagenesis only on F208.

[Comment 3]

What is the reason for performing experiments with chimeric AL domains? Since the identified FAAL domain has been shown to be responsible for the specificity, a mutational analysis of the acyl chain binding pocket could have done the job. The authors may want explain why they adopted such a strategy!

>[Response]

I really appreciate your valuable suggestion. We had tried to predict the amino acid residues by modeling the structure of mycAL or ituAL, which is exactly as you said. However, their identity toward structure solved FAALs, such as 3KXW, 5HM3, 3E53 or 3PBK, are less than 30% and we considered that modeling based on these structure is not reliable for predicting the amino acid residues, which are responsible for the acyl chain length.

Based on the above, the following text was added in lines 166-173.

“The AL domain of iturin family lipopeptide synthetases is classified as fatty acyl-AMP ligase (FAAL). To determine the amino acid residues, the structural modeling is usually employed, however this strategy is not favourable because MycAL / ItuAL is less than 30% identical to the structure solved FAALs. Therefore, we experimentally verified all amino acid residues which are different between MycAL and ItuAL.

To narrow down the regions affecting the relative abundance of mycosubtilin homologues, we constructed mutant strains harbouring MycAL and ItuAL chimaeras.”

[Comment 4]

Why was the in vitro activity of ALs on C-17 and C-19 fatty acids not tested for the F208G mutant, despite their higher antifungal activity?

>[Response]

I apologize for confusion. In iturin synthesis by iturin synthetase, fatty acids (C_n) are taken by AL domain at first. Then β -keto acyl synthetase (KS) domain elongates two carbons and convert fatty acids into C_{n+2} . Therefore, we focused on C15 fatty acids (n-, iso, anteiso) for enzymatic assays, because C15 fatty acids are substrates for C17-iturins.

To avoid misunderstanding, the following text was added in lines 306-309.

“In iturin family lipopeptide synthetases, fatty acids are taken up by the AL domain and then elongated by two carbons by adjacent KS domains (Fig. 1). This means that the C12 and C15 fatty acids incorporated as substrates correspond to the products with the C14 and C17 fatty acid moieties, respectively.”

Other revisions

I corrected several grammatical mistakes and representations according to the guideline of the journal. These corrections are also written in red colour.